# Patterns and Predictors of Sitting among Women from Disad-Vantaged Neighbourhoods over Time: A 5-Year Prospective Cohort Study

**DOI:** 10.3390/ijerph18094625

**Published:** 2021-04-27

**Authors:** Minakshi Nayak, Karen Wills, Megan Teychenne, Jo Salmon, Verity Cleland

**Affiliations:** 1Menzies Institute for Medical Research, University of Tasmania, Private Bag 23, Hobart, TAS 7000, Australia; minakshi.nayak@utas.edu.au (M.N.); karen.wills@utas.edu.au (K.W.); megan.teychenne@deakin.edu.au (M.T.); 2Institute for Physical Activity and Nutrition (IPAN), School of Exercise and Nutrition Sciences, Deakin University, Geelong, VIC 3220, Australia; jo.salmon@deakin.edu.au

**Keywords:** sitting time, sedentary behaviour, women, low socioeconomic position

## Abstract

*Background*: Our aim was to describe patterns of sitting over time and determine the sociodemographic predictors of sitting over time among women living in socioeconomically disadvantaged neighbourhoods. *Methods:* Women age between 18 and 45 years (mean = 34.4 ±8.1, *n* = 4349) reported their sitting time, sociodemographic (e.g., age), and health (e.g., body mass index) three times over 5 years. Linear mixed modelling was used to determine the predictors of change in sitting over time, adjusting for covariates. *Results:* Mean baseline sitting time was 40.9 h/week, decreasing to 40.1 h/week over five years. Greater sitting time was reported in participants ≤25 years of age, living with obesity, living in urban areas, self-reported poor/fair health, working full-time, with higher education, never married and with no children. Annually, the average sitting time decreased by 0.4 h/week (95% CI; −0.7 to −0.05) in women working full-time but increased by 0.1 h/week (95% CI; −0.2 to 0.6) who were not working. Similarly, annual sitting time decreased by 0.6 h/week (95% CI; −0.2 to 1.3) in women with no children but increased by 0.4 h/week (95% CI; −0.2 to 0.5) and 0.9 h/week (95% CI; 0.3 to 1.3) among those with two and three/more children, respectively. *Conclusion:* Among disadvantaged women, those not working and with two or more children may be at particular risk for increased sitting time and warrant further attention.

## 1. Introduction

Sitting is a sedentary behaviour (SB) where energy expenditure is ≤1.5 metabolic equivalent (METs) [1] and is increasingly recognised as an important public health issue [2]. High levels of SB are associated with a greater risk of mortality, obesity, type-II diabetes, cardiovascular diseases, depression, and anxiety [3,4]. These associations remain after adjusting for physical activity [5]. These diseases and conditions are strongly socioeconomically patterned, with those of lower socioeconomic position (SEP) experiencing them at higher rates than those of higher SEP [6]. The lifestyle behaviours (e.g., physical inactivity) related to these diseases are all also socioeconomically patterned [7] Research suggests that SB may also be socioeconomically patterned [8], and hence understanding patterns of sitting time specifically among socioeconomically disadvantaged groups is warranted.

Additionally, understanding the factors associated with sitting time may help identify population groups at increased risk of chronic disease and identify potential targets for intervention. A range of factors including age [9,10,11], obesity [10,12], general health status [13], marital status [10], occupation [10,11,14], area of residence (rural/urban) [15] and sex [9,16,17] have been identified as correlates of sitting time. Other socioeconomic indicators such as education, employment, and income have also been associated with sitting time, although evidence is less consistent. For example, studies have found positive associations [18,19,20] (those of higher SEP spent 60–70 min/day more time on sitting than lower SEP) as well as inverse associations (those of lower SEP spent (70 to 80%) more time on sitting than higher SEP during weekend and leisure) [17] between indicators of SEP and sitting time. Although studies show evidence of higher sitting time in men [16,17], the negative impact of sedentary time (e.g., cardiovascular death) is more pronounced in women [21]. Given inconsistent findings on SEP, there have been calls for further research to examine the longitudinal patterns and predictors of sitting time [8] so that it can be established whether there are groups at high risk that could be targeted.

Among women from socioeconomically disadvantaged neighbourhoods, this study aimed to:Describe patterns of time spent in sitting over time;Identify baseline sociodemographic predictors of sitting time patterns over time.

## 2. Methods

Data were obtained from a postal survey conducted for the Resilience for Eating and Activity Despite Inequality (READI) study, approved by the Deakin University Human Research Ethics Committee (ID-2006-091), Australia. Participants provided written consent.

### 2.1. Participants

READI was a prospective cohort study of women aged 18–45 years living in socioeconomically disadvantaged neighbourhoods (urban and rural suburbs) in the state of Victoria, Australia, described in detail elsewhere [22]. Briefly, 40 rural and 40 urban low socioeconomic neighbourhoods were randomly selected for sampling, with 150 women from each neighbourhood randomly selected from the electoral roll and invited to participate by mail. In 2007–2008 (baseline/T1), 2010–2011 (first follow-up/T2) and 2012–2013 (second follow-up/T3), 4349, 1912 and 1560 women participated, respectively (Figure 1).

### 2.2. Measures

#### 2.2.1. Outcome: Weekly Sitting Time

Total time spent sitting per week was reported via the long version of the International Physical Activity Questionnaire (IPAQ-L) [23] at each of the three time points. Sitting items from IPAQ-L have shown good test-retest reliability (Spearman rho values above 0.70) and acceptable validity (Spearman’s correlation coefficient of 0.30 with accelerometer data) [23,24]. Participants reported usual (min/day) time spent sitting in the past week, both weekday and weekend day. Total weekly sitting time was calculated by multiplying the duration of sitting time on weekdays by five and weekend days by two, then adding these two values. As per the IPAQ protocol [25], which assumes 8 h sleep in a 24 h period [26], weekly sitting time values greater than 16 h/day were considered implausible (T1 = 467, T2 = 193, T3 = 63) and treated as missing.

#### 2.2.2. Exposures: Baseline Sociodemographic and Health Characteristics

Participants reported age, weight, height, educational qualification (low = less than Year 12, medium = Year 12, trade/certificate/diploma, high = tertiary), employment status (full-time, part-time, not working), weekly income (no income, $1–699, $700 AUD and above, don’t know/want to answer), marital status (married/de facto relationship, widow/separated/divorced, never married), number of children (<18 years) living at home (none, one, two, three or more), smoking status (never smoked, used to smoke, occasionally, current smoker). Participants on self-reported health status were asked to respond the following “would you say your health is: excellent, very good, good, fair, poor”. Age was categorised (18–25 years, 26–35 years, 36–45 years) as per previous studies that have reported being young (18–25 years) was associated with higher sitting time compared to older [9,10,11]. Body mass index (BMI) (kg/m^2^) was calculated from self-reported weight (kg) and height (m), and classified as normal (BMI 18–25 kg/m^2^), overweight (BMI 26–30 kg/m^2^) or obese (BMI > 30 kg/m^2^) [27]. Urban-rural status was allocated during sampling.

### 2.3. Statistical Analysis

Participant characteristics were summarised by frequency (*n*) and percentage (%) for categorical data and by the mean and standard deviation for continuous data. Baseline characteristics of participants lost to follow-up were compared to the participants who completed surveys at all three time points because the difference between the samples could affect the internal validity of the study results [28].

The associations of sociodemographic factors with changes in hours of sitting time per week over five years were estimated using linear mixed models, which account for correlated observations due to repeated measurements on individuals. Separate models were fit for each baseline sociodemographic and health exposure (BMI and self-reported health status) variable. Each model included fixed effects terms for time (years since baseline) and sociodemographic and health exposure variables and an interaction term for these two factors (e.g., time*age). The interaction term estimates the additional change in sitting time per year associated with each category of the exposure variable. A random intercept was specified for each participant to account for individual differences in baseline sitting time, and time was also specified as a random effect to allow the effect of time to vary among individuals. The correlation between the repeated measurements over time was modelled using an exponential covariance structure with an unstructured covariance matrix for the random effect. All models were optimised using maximum likelihood estimation. A likelihood-ratio test was used to compare the models with and without a random effect of time. As a random effect for time significantly improved all models’ fit, it was retained throughout. Adjusted models used the criteria of a 10% change in estimates to identify confounders for each individual model [29].

Model estimates are presented as baseline differences for categories of exposure variables compared with the reference category, overall average change per year, and additional annual change for each category of the exposure compared with the reference category. The estimated annual change in sitting time for each category of the exposure was calculated by adding the overall average change per year and the category-specific additional change per year.

A number of observations missing for both exposure and outcome were described in Figure 1. Missing data were imputed using the method of chained equations, assuming data were missing at random [30]. Multiple imputation is a statistical technique, which involves replacing each missing cell with multiple values based on information from the observed variables of the dataset; as a result, it generates an unbiased distribution of missing data [31]. The imputation model included baseline variables for age, BMI, smoking, health status, area of residence, education, employment, income, marital status, number of children, and sitting time at all three time points. Interaction variables were generated using dummy variables for each exposure and a combination of time before imputation to avoid bias estimation [32]. Chained equations imputation models were run separately for each analysis model due to the complexity of treating levels of categorical variables and their interaction as dummy variables. Fifty imputed datasets were created, and the estimates from the multiple imputed datasets were combined into an overall estimate using Rubin’s rules [33].

Data analysis was conducted in 2019 using STATA version 15 (Stata Corp, College Station, TX, USA) with *p* < 0.05 considered statistically significant.

### 2.4. Sensitivity Analyses

The first sensitivity analysis examined the impact of missing data by conducting a complete case analysis where only data used from participants who responded to all three surveys.

The second sensitivity analysis examined the potential impact of high/implausible sitting time (values over 960 min/day at T1 = 11%, T2 = 10% and T3 = 4.3%). We hypothesised that participants with high values may have reported on total weekday and weekend estimates rather than on single-day estimates of weekday and weekend. This hypothesis was somewhat supported as a high proportion of implausible values were divisible by five (weekdays) and two (weekends). Observations that remained higher than 960 min/day after dividing by five and two were truncated to 960 min/day.

The results from both sensitivity analyses were compared to the main results to determine whether there were changes in the magnitude or direction of the estimated effect or in statistical inference.

## 3. Results

### 3.1. Characteristics of the Sample

Mean sitting time was 40.9 h/week (2454 min/week) at T1, decreasing to 40.3 h/week (2418 min/week) (T2) and 40.1 h/week (2406 min/week) (T3). Table 1 presents the sociodemographic and health characteristics of the sample.

### 3.2. Loss to Follow-Up

Compared to women who participated only at T1 or only T1 and T2, women remaining in the study were older, non-smokers, had better health, higher education, higher incomes, living in rural areas, working part-time, married, and had children (Appendix A).

### 3.3. Observed Sitting Time Patterns for Demographic and Health-Related Factors

Mean sitting time was typically highest among youngest participants, living with obesity, living in urban areas, with poor/fair health, higher education, working full-time, higher income, never married, and with no children (Table 2).

### 3.4. Sociodemographic and Health-Related Factors Associated with Sitting Time

In adjusted models, women living in rural areas sat 3.5 h/week less at baseline than women living in urban areas, women working part-time and those not working sat 5.1 and 7.6 h less per week at baseline than women working full-time, women with one, two, or three or more children sat 3.7, 5.5 and 7.4 h/week less, respectively, at baseline than women with no children (Table 3). Higher baseline sitting time was found for women with higher BMI (2.0–3.3 h/week), higher incomes (≥ $700 AUD/week as compared to nil, 6.3–14.4 h/week), and those never married (4.8 h/week). Despite these baseline differences, apparent, no longitudinal associations were observed for baseline age, BMI, area of residence, health, education, income, or marital status. That is, no average change per year in sitting time for these variables. However, longitudinal associations were observed for employment status and number of children.

In regard to the annual change in sitting time in the adjusted model for employment status, annual sitting time decreased on average by 0.4 h/week (95% CI: −0.7 to −0.05), or 24 min/week. However, women who were not working at baseline increased sitting time by 0.1 h/week (95% CI: −0.2 to 0.6), or 6 min/week per year (average change in sitting time in all employment groups (−0.4) + average change in sitting time among those not working (0.5) = overall annual increase (0.1 h)).

In regard to the annual change in sitting time in the adjusted model for the number of children, annually sitting time decreased on average by 0.6 h/week (95% CI: −0.2 to 1.3), or 36 min/week. However, women with two children at baseline increased sitting time by 0.4 h/week (95% CI: −0.2 to 0.5) or 24 min/week per year (average change in all children groups (−0.6) + average change in sitting time among women with two children (1.0) = overall annual increase (0.4 h/week)). Similarly, women with three or more children at baseline increased sitting time by 0.9 h/week (95% CI: 0.3 to 1.3) or 54 min/week per year (average change in all children groups (−0.6) + change in sitting time among women with three or more children (1.5) = overall annual increase (0.9 h/week)).

### 3.5. Sensitivity Analyses

Sensitivity analyses using complete case data indicated small changes in estimates (difference in beta values range between 0 and 1.2), but results remained consistent, and no changes in inferences were apparent (Appendix A). Sensitivity analyses using the truncation method for implausible sitting time values produced some differences in estimated associations (Appendix A). Effect size at baseline and average change over time for every variable were stronger, suggesting this procedure may overestimate the strength of association. Most inferences remained the same, although average change over time was no longer significant for age and area of residence.

## 4. Discussion

This study described patterns of sitting time over time and identified sociodemographic and health factors associated with changes in sitting over time among young to middle-aged women living in socioeconomically disadvantaged neighbourhoods. Based on the finding of adjusted models, in this sample, sitting time declined over time, identical to previously reported studies on European [34] and Australian adults [15]. Sitting time was higher among those who were young (18–25 years), with higher BMI, living in an urban area, with medium or higher education, with higher incomes, in full-time employment, and separated/divorced/widowed or never married, consistent with previous studies [9,10,11,14,15,18,35]. Sitting time was lower among women with children is supported by other studies [13,15,18]. Employment status and number of children emerged as predictors of changes in sitting time over time. Average time spent in sitting decreased over time among women working full-time and those with no children but increased among those not working and with two or more children at baseline.

### 4.1. Patterns and Predictors of Sitting

Our study found that younger women (18–24 years) sat more than older women (35–45 years), consistent with the Australian Longitudinal Study on Women Health (ALSWH) study [14]. In contrast, one study reported being older was related to higher sitting time among women [13], although unlike our sample of low SEP women, participants in that study were university employees, so comparisons should be made with caution. Our finding that younger women spent more time sitting could be explained by greater use of technology [36] or having more sedentary occupations [10,36]. Further, our results on women with a higher BMI sat more than normal BMI is consistent with other studies [10,12]. There has been debate as to whether increased weight leads to higher sitting time or vice versa. Although two previous prospective studies did not establish bidirectional associations [37,38], a study by Ekuland et al. suggested that BMI predicted sitting time [37]. It may be that women living with overweight or obesity prefer more sedentary occupations or leisure pursuits. The current study found that urban women sat more than rural women. Similar to prior Australian studies [15,18] and a Chilean study [9], despite these studies having a higher proportion of participants from urban neighbourhoods [9,15,18] and high SEP [18]. Our results suggest an existence of a rural-urban discrepancy in sitting time among those of low SEP as well.

This study describes the longitudinal patterns of total sitting time among women living in socioeconomically disadvantaged neighbourhoods, which extends previous research that has typically only assessed socioeconomic patterns of SB based on income and education [9,10,14,35,39,40]. Women with higher education and incomes in disadvantaged areas may be more likely to have more sedentary occupations. Further, women with full-time employment may have a greater opportunity to engage in more prolonged work-related sitting, possibly contributing to higher overall sitting time. However, our study demonstrated that women not working at baseline increased their sitting time more than those working full-time. This contrasts with findings from the ALSWH study, which reported work-related factors such as higher occupational status (professionals) and longer hours worked were associated with greater sitting time [15]. Higher sitting time among full-time workers in the ALSWH may be explained by the fact that participants reported professional occupations (often involve desk-based activities). In contrast, it has been suggested that unemployed adults engage in more unhealthy lifestyle behaviour due to loss of motivation, low self-efficacy or stress [41], thus may reallocate time for leisure activities that are generally passive such as watching television [42]. It could also be that women are not working at baseline obtained jobs over the study period. Given the inconsistencies between studies regarding the association between education, income, employment status, and sitting time, further studies are required to disentangle these relationships.

The current study showed that single women sat more than married women. This finding is consistent with existing hypotheses where marriage appears to be protective against unhealthy behaviours [43]. Similarly, this hypothesis might explain the inverse relationship between sedentary time and the presence of children at home. Having children involves physical domestic responsibilities (e.g., care) and potential engagement in different indoor and outdoor activities (e.g., playing). Together, our cross-sectional findings indicate that family commitments lower the sitting time among disadvantaged women. However, we found that over time women with two or more children at baseline increased their time spent sitting compared to those with no children. In contrast, a previous longitudinal study showed that women with any number of children sat less than women with no children [15]. In the current study, around 40% of women reported not having children, and 26% were not married at baseline. During the study period, these participants may have married or had children, which may have resulted in decreased sitting time. Similarly, around half of the participants were between the ages of 36–45 years and most reported having two or three or more children at baseline. As mid-aged women are more likely to have older children with fewer physical demands than required for younger children (e.g., less time in childcaring activities), this may result in increased leisure-time sitting, such as watching TV. Alternatively, mothers with two or more children may spend more time transporting their children in cars (e.g., school, extra-curricular activities) and sitting watching them participate in various activities or may have greater work hours. Therefore, further study is warranted to investigate whether the arrival of a newborn or subsequent child change sitting time in mothers. In addition, investigating different domains of sitting time, such as leisure-time versus transport or occupational, is recommended in order to gain a better understanding of the potential explanatory factors for these associations.

### 4.2. Strength and Limitations

The strengths of this study included its prospective design with data collected at three time points over five years in a large population-based sample of both urban and rural-dwelling low SEP women. It is one of the first studies internationally to describe sitting time patterns and predictors of sitting time patterns over time and among women living in socioeconomically disadvantaged areas, a group at high risk for chronic disease. Limitations of this study include the self-reported data. IPAQ-L sitting items have good test-retest reliability over 4 weeks [24], but it is not clear how sensitive this instrument is to detecting change over five years. Further development of reliable measures of sitting, capturing domain-specific behaviour is needed. Secondly, a large proportion of participants were lost to follow-up (loss rate = 56%). To address potential biases related to loss to follow-up, we used multiple imputation to estimate the missing values of the participants, potentially reducing attrition bias. This study included only women between the age of 18–45 years living in socioeconomically disadvantaged neighbourhoods, so results may not be generalisable to men, older women, or women living in less socioeconomically disadvantaged neighbourhoods.

## 5. Conclusions

This study identified two population subgroups (not working and with two or more children) of women from socioeconomically disadvantaged neighbourhoods at the greatest risk of increasing sitting time. Given the detrimental health effects of sitting time, these groups should be targeted in future interventions for reducing SB. Furthermore, for a better understanding of potential factors for these associations and inform the development of SB interventions for these groups, future studies need to investigate the predictors associated with a change in domain (e.g., leisure, transport, work) and context-specific (e.g., TV viewing, computer) sitting behaviours.

## Figures and Tables

**Figure 1 ijerph-18-04625-f001:**
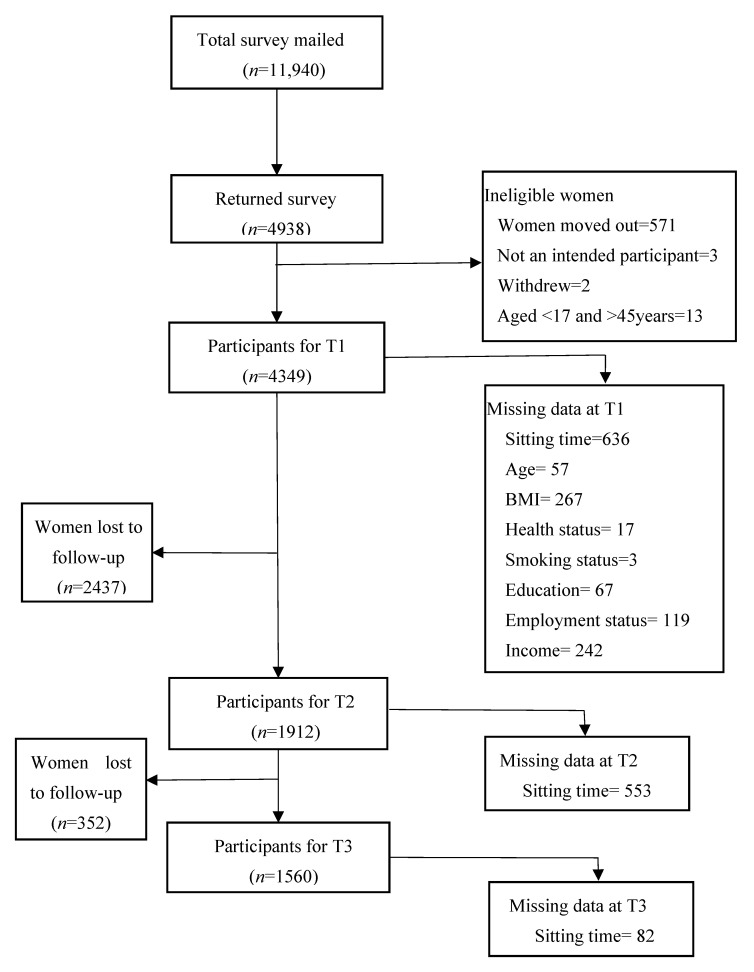
Flow chart of READI participants.

**Table 1 ijerph-18-04625-t001:** Sociodemographic and health characteristics of participants in the READI study.

Variables	T1 (*n* = 4349)	T2 (*n* = 1912)	T3 (*n* = 1560)
Age (years) (mean (SD))	34.4 (8.1)	39.0 (7.7)	41.2 (7.6)
Age category *n* (%)			
18–25 years	741 (17.4)	121 (6.6)	42 (2.97)
26–35 years	1299 (30.1)	422 (23.1)	307 (20.3)
36–45 years	2252 (52.4)	1286 (70.3)	1161 (76.8)
BMI (kg/m^2^) (mean, SD)	26.05 (6.0)	26.8 (6.3)	26.7 (6.3)
BMI category (kg/m^2^) *n* (%)			
Normal (18–25)	2153 (52.7)	801 (43.2)	682 (44.7)
Overweight (2630)	1037 (25.4)	479 (25.9)	323 (25.9)
Obese (>30)	892 (21.8)	572 (30.9)	439 (29.4)
General health *n* (%)			
Excellent	392 (9.1)	205 (11.2)	176 (11.7)
Very good	1508 (34.8)	706 (38.2)	581 (38.6)
Good	1799 (41.5)	730 (39.5)	571 (39.7)
Poor or fair	631 (14.6)	207 (11.2)	149 (9.9)
Smoking *n* (%)			
Never smoked	2183 (50.2)	955 (51.7)	817 (53.5)
Used to smoke	1066 (24.5)	542 (29.4)	440 (29.8)
Smoke occasionally	411 (9.5)	125 (6.7)	97 (6.5)
Current smoker	684 (15.8)	225 (12.1)	153 (10.2)
Area of residence *n* (%)			
Urban	2016 (46.4)	669 (39.6)	587 (39.7)
Rural	2331 (53.6)	1020 (60.4)	891 (60.3)
Marital Status *n* (%)			
Married	2829 (65.5)	1384 (73.3)	1122 (74.6)
Widowed/separated/divorced	370 (8.5)	83 (4.5)	67 (4.4)
Never married	1122 (26.0)	413 (22.4)	315 (20.9)
Education level *n* (%)			
Low (did not complete high school)	946 (22.1)	372 (20.6)	266 (17.7)
Medium (completed high school)	2216 (51.7)	899 (48.6)	713 (47.4)
High (completed tertiary education)	1120 (26.2)	580 (30.9)	525 (34.9)
Employment status *n* (%)			
Working full-time	1613 (38.1)	739 (38.9)	610 (39.6)
Working part-time	1245 (29.4)	652 (34.4)	576 (37.4)
Not working	1372 (32.4)	505 (26.6)	354 (22.9)
Household gross income *n* (%)			
No income	294 (7.4)	125 (7.0)	98 (6.6)
$1–699 AUD/week	2235 (56.5)	885 (49.8)	667 (45.2)
$700–1499 AUD/week	880 (22.2)	485 (27.3)	465 (31.5)
$1500 AUD and above /week	99 (2.5)	103 (5.8)	115 (7.8)
Do not know/want to answer	448 (11.3)	179 (10.1)	131 (8.9)
Number of children *n* (%)			
None	1678 (39.4)	63 (5.1)	99 (9.7)
One	787 (18.5)	327 (26.3)	270 (26.4)
Two	1086 (25.5)	511 (41.2)	403 (39.4)
Three or more	713 (16.7)	340 (27.4)	252 (24.6)

Abbreviation: BMI—body mass index, SD—standard deviation, *n* (%)—number(percentage), T1 = 2007–2008, T2 = 2010–2011, T3 = 2012–2013.

**Table 2 ijerph-18-04625-t002:** Mean (SD) sitting time (h/week) and sociodemographic and health characteristics in the READI study.

Variables		Sitting Time (h/week)	
	T1 (2007–2008)	T2 (2010–2011)	T3 (2012–2013)
Age (years)			
18–25	44.0 (20.9)	41.2 (20.2)	46.3 (18.3)
26–35	42.6 (22.0)	41.7 (21.7)	39.9 (20.1)
36–45	39.2 (29.2)	39.9 (21.4)	40.0 (20.6)
BMI (kg/m^2^)			
Normal (18–25)	39.7 (21.4)	38.8 (19.0)	39.2 (19.9)
Overweight (26–30)	41.2 (21.3)	41.9 (20.8)	40.0 (20.1)
Obese (>30)	43.1 (21.9)	43.3 (21.4)	42.1 (20.2)
Area of residence			
Urban	43.7 (22.0)	43.1 (21.3)	42.3 (20.9)
Rural	38.6 (20.7)	38.6 (21.3)	38.6 (20.1)
General health			
Excellent	38.4 (20.9)	38.0 (21.3)	38.4 (20.0)
Very good	40.9 (21.4)	39.1 (20.9)	39.3 (19.6)
Good	41.4 (21.6)	41.2 (21.2)	40.1 (20.4)
Poor/fair	42.6 (21.3)	44.9 (23.7)	46.3 (23.5)
Smoking			
Never smoked	40.8 (21.3)	40.1 (21.3)	40.3 (21.2)
Used to smoke	40.9 (23.2)	40.3 (20.9)	40.7 (19.3)
Smoke occasionally	41.0 (21.0)	41.2 (23.7)	40.9 (19.2)
Smoke regularly	41.4 (22.3)	41.8 (22.1)	37.2 (20.1)
Education			
Low	38.9 (21.5)	40.8 (22.8)	39.1 (19.8)
Medium	41.4 (22.0)	40.7 (19.3)	39.8 (20.4)
High	41.9 (20.5)	40.0 (20.3)	41.2 (21.0)
Employment			
Working full-time	45.9 (21.2)	43.2 (21.0)	44.1 (20.2)
Working part-time	38.8 (21.6)	38.6 (20.8)	37.3 (19.2)
Not working	37.2 (20.8)	38.5 (20.3)	37.8 (21.9)
Average gross household income			
None	36.0 (21.6)	37.9 (21.8)	37.4 (21.7)
$1–699 AUD/week	39.8 (21.4)	39.2 (21.5)	37.0 (18.7)
$700–1499 AUD/week	45.8 (20.6)	44.1 (20.2)	44.1 (20.8)
$1500 AUD and above/week	54.1 (20.9)	45.1 (20.6)	48.6 (20.6)
Don’t know/want to answer	40.4 (21.8)	38.8 (22.2)	36.7 (20.3)
Marital status			
Married	39.4 (21.2)	39.3 (21.4)	39.5 (20.5)
Separated/divorced/widowed	39.3 (21.7)	38.1 (20.2)	35.4 (19.4)
Never married	46.8 (21.3)	44.8 (21.4)	45.0 (20.4)
Number of children			
None	46.8 (20.9)	47.2 (21.9)	41.8 (22.7)
One	39.7 (22.0)	38.8 (21.4)	37.6 (19.6)
Two	37.3 (21.1)	37.2 (21.4)	37.3 (19.9)
Three or more	34.7 (19.5)	34.9 (19.8)	35.1 (19.5)

Abbreviations: BMI—body mass index, SD—standard deviation, T1 = 2007–2008, T2 = 2010–2011, T3 = 2012–2013.

**Table 3 ijerph-18-04625-t003:** Change in sitting time over five years by sociodemographic and health characteristics.

		Unadjusted Model			Adjusted Model		
	Variables	β-Coefficient (95% CI)		*p*-Value	β-Coefficient (95% CI)		*p*-Value
Age category ^a^	Baseline difference						
	18–25 years	Ref			Ref		
	26–35 years	−1.3	(−3.9, 0.8)	0.216	−0.4	(−2.5, 1.6)	0.694
	36–45 years	−4.5	(−6.4, −2.6)	**<0.001**	−1.3	(−3.2, −0.7)	0.214
	Average change per year	−0.3	(−1.0, 0.3)	0.312	−0.3	(−0.9, 0.3)	0.328
	18–25 years						
	26–35 years	−0.3	(−1.1, 0.4)	0.382	−0.2	(−1.0, 0.5)	0.499
	36–45 years	0.5	(−0.2, 1.2)	0.164	0.5	(−0.2, 1.1)	0.166
BMI ^b^	Baseline difference						
	Normal	Ref			Ref		
	Overweight (26–30)	1.8	(0.1, 3.4)	**0.038**	2.0	(0.6, 3.8)	**0.008**
	Obese (>30)	3.3	(1.6, 5.1)	**<0.001**	3.3	(1.5, 5.1)	**<0.001**
	Average change per year	0.0	(−0.3, 0.3)	0.963	0.1	(−0.2, 0.4)	0.495
	Normal						
	Overweight	−0.3	(−0.8, 0.3)	0.302	−0.3	(−0.8, 0.2)	0.194
	Obese (>30)	−0.4	(−0.9, 0.2)	0.178	−0.4	(−1.0, 0.1)	0.131
Area of Residence ^c^	Baseline difference						
	Urban	Ref			Ref		
	Rural	−5.1	(−6.4, −3.7)	**<0.001**	−3.5	(−4.9, −2.2)	**<0.001**
	Average change per year	−0.2	(−0.6, 0.1)	0.174	−0.2	(−0.5, 0.1)	0.217
	Urban						
	Rural	0.2	(−0.2, 0.6)	0.322	0.2	(−0.2, 0.7)	0.279
General	Baseline difference						
Health ^d^	Excellent	Ref			Ref		
	Very good	1.7	(−0.7, 4.3)	0.174	0.8	(−1.6, 3.2)	0.541
	Good	2.8	(0.3, 5.2)	**0.020**	1.1	(−1.3, 3.6)	0.350
	Poor/fair	4.5	(1.6, 7.3)	**0.002**	1.7	(−1.2, 4.7)	0.238
	Average change per year	−0.1	(−0.8, 0.5)	0.678	−0.1	(−0.6, 0.7)	0.961
	Excellent						
	Very good	0.1	(−0.7, 0.8)	0.833	0.0	(−0.7, 0.8)	0.900
	Good	−0.2	(−0.9, 0.5)	0.580	−0.3	(−1.0, 0.5)	0.475
	Poor/fair	0.4	(−0.5, 1.2)	0.363	0.3	(−0.6, 1.1)	0.488
Smoking	Baseline difference						
Status ^e^	No smoking	Ref			Ref		
	Used to smoke	0.0	(−1.6, 1.7)	0.956	1.0	(−0.6, 2.7)	0.213
	Smoke occasionally	0.1	(−2.3, 2.5)	0.958	−0.8	(−3.1, 1.6)	0.511
	Smoke regularly	1.1	(−0.8, 3.1)	0.256	1.1	(−0.9, 3.1)	0.297
	Average change per year	−0.1	(−0.4, 0.2)	0.410	0.0	(−0.3, 0.3)	0.962
	No smoking						
	Used to smoke	0.0	(−0.5, 0.5)	0.918	0.0	(−0.5, 0.5)	0.896
	Smoke occasionally	0.3	(−0.5, 1.1)	0.407	0.3	(−0.5, 1.1)	0.488
	Smoke regularly	−0.5	(−1.1, 0.2)	0.170	−0.5	(−1.2, 0.1)	0.102
Education	Baseline difference						
Status ^f^	Low	Ref			Ref		
	Medium	1.9	(0.1, 3.6)	**0.035**	0.2	(−1.5, 1.9)	0.813
	High	2.2	(0.3, 4.2)	**0.025**	0.0	(−1.9, 2.0)	0.986
	Average change per year	0.0	(−0.5, 0.4)	0.914	0.0	(−0.5, 0.4)	0.891
	Low						
	Medium	−0.2	(−0.8, 0.3)	0.380	−0.1	(−0.7, 0.4)	0.617
	High	0.0	(−0.6, 0.5)	0.913	0.1	(−0.5, 0.7)	0.781
Employment	Baseline difference						
Status ^g^	Full-time	Ref			Ref		
	Working part-time	−6.4	(−8.1, −4.8)	**<0.001**	−5.1	(−6.7, −3.4)	**<0.001**
	Not working	−8.4	(−10.0, −6.8)	**<0.001**	−7.6	(−9.2, −6.0)	**<0.001**
	Average change per year	−0.5	(−0.8, −0.1)	**0.006**	−0.4	(−0.7, −0.05)	**0.023**
	Full-time						
	Working part-time	0.4	(−0.1, 0.9)	0.083	0.4	(−0.04, 1.01)	0.146
	Not working	0.6	(0.05, 1.1)	**0.026**	0.5	(0.05, 1.1)	**0.031**
Average Gross Income ^h^	Baseline difference						
	Nil	Ref			Ref		
	$1–699 AUD/week	2.9	(0.5, 5.3)	**0.017**	1.8	(−0.6, 4.2)	0.503
	$700–1499 AUD/week	8.5	(5.8, 11.3)	**<0.001**	6.3	(3.5, 9.0)	**0.002**
	$1500 AUD and above/week	16.5	(11.6, 21.3)	**<0.001**	14.4	(9.7, 19.2)	**<0.001**
	Don’t know/want to answer	3.2	(0.1, 6.3)	**0.042**	1.6	(−1.4, 4.7)	0.736
	Average change per year	−0.3	(−0.9, 0.4)	0.386	−0.1	(−0.7, 0.6)	0.960
	Nil						
	$1–699 AUD/week	0.3	(−0.4, 1.0)	0.414	0.2	(−0.5, 0.9)	0.633
	$700–1499 AUD/week	−0.1	(−0.9, 0.7)	0.744	−0.3	(−1.1, 0.5)	0.771
	$1500 AUD and above/week	−1.2	(−2.6, 0.2)	0.104	−1.3	(−2.7, 0.1)	0.267
	Don’t know/want to answer	−0.5	(−1.4, 0.6)	0.343	−0.7	(−1.7, 0.3)	0.186
Marital Status ^i^	Baseline difference						
	Married	Ref			Ref		
	Separated/divorced/widowed	0.1	(−2.3, 2.6)	0.915	−0.3	(−2.7, 2.1)	0.819
	Never married	7.4	(5.8, 9.0)	**<0.001**	4.8	(2.9, 6.5)	**<0.001**
	Average change per year	−0.1	(−0.3, 0.1)	0.382	−0.1	(−0.3, 0.2)	0.542
	Married						
	Separated/divorced/widowed	0.4	(−0.4, 1.2)	0.291	0.4	(−0.4, 1.2)	0.357
	Never married	−0.1	(−0.6, 0.4)	0.696	−0.2	(−0.8, 0.3)	0.503
Number of Children ^j^	Baseline difference						
	No children	Ref			Ref		
	One	−6.3	(−8.5, −4.7)	**<0.001**	−3.7	(−5.7, −1.7)	**<0.001**
	Two	−9.1	(−10.8, −7.4)	**<0.001**	−5.5	(−7.3, −3.6)	**<0.001**
	Three or more	−11.8	(−13.7, −9.8)	**<0.001**	−7.4	(−9.5, −5.3)	**<0.001**
	Average change per year	−0.7	(−1.1, −0.4)	**<0.001**	−0.6	(−1.1, −0.3)	**0.001**
	No children						
	One	0.5	(−0.1, 1.1)	0.114	0.5	(−0.1, 1.1)	0.102
	Two	0.9	(0.3, 1.4)	**0.001**	1.0	(0.4, 1.5)	**<0.001**
	Three or more	1.5	(0.9, 2.1)	**<0.001**	1.5	(0.9, 2.1)	**<0.001**

Abbreviations: Ref—reference, BMI—body mass index, CI—confidence interval. Bold text indicates statistically significant result with a p-value ≤ 0.05. Each model separately adjusted as follows: ^a^ Age category for BMI, area of residence, and number of children. ^b^ BMI category for age, area of residence, health status, education, employment, marital status, and number of kids. ^c^ Area of residence for BMI, marital status, and number of children. ^d^ Health status for age, BMI, area of residence, education, employment, marital status, and number of children. ^e^ Smoking status for age, BMI, area of residence, education, health status, marital status, and number of children. ^f^ Employment status for area of residence and marital status. ^g^ Education status for BMI, area of residence, employment status, health status, marital status, and number of children. ^h^ Income status for age, BMI, health status, marital status, and number of children. ^i^ Marital status for age, area of residence and number of children. ^j^ Number of children for area of residence, employment status, and marital status.

## Data Availability

The data used in this study are available on request from the corresponding author.

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
