# Peer review of "Patterns and Predictors of Sitting among Women from Disad-Vantaged Neighbourhoods over Time: A 5-Year Prospective Cohort Study"

_ijerph, 2021, doi:10.3390/ijerph18094625_

Round 1

Reviewer 1 Report

Overall comment

This was a prospective cohort study (3 time points of assessment) aiming to assess how time spent in sitting changes over a 5-year period in socioeconomically disadvantaged neighborhoods women (18– 45 yrs) in Australia and identify baseline predictors of time spent in sitting over that period. It is a well written manuscript, with appropriate Introduction/background, Methods, Results, Discussion and Conclusions. I have a few suggestions to improve the overall quality of the reporting, divided in “throughout the text” and “specific”, i.e., on each section of the manuscript.

“Throughout the text”:

– The authors omit in most parts of the text the attribute of the sitting term which is sitting time or time spent in sitting. This is relevant because the authors claim they will study sitting patterns, but in fact they only studied overall time spent in sitting, as a measure of sedentary behavior. Sitting patterns could be sitting posture, sitting time or posture while working, sitting time with legs crossed while watching TV, sitting lumbopelvic angle while driving in urban areas, etc. IPAQ-L cannot differentiate between those, hence, “sitting patterns” is not conceptually correct (In this study) and should be amended to sitting time or time spent in sitting whenever applicable.

– I usually don’t recommend changing, in most revisions, the decimals in central tendency and dispersion descriptive statistics in variables related with time that have a particular subdivision system (years and months; hours and minutes), e.g., Age = 54.6 ± 14.3 years old, but because the focus of this research is time spent in sitting, I found that reporting time spent in sitting as 40.2 hours/week can be exhausting for readers to convert constantly the decimals to minutes in order to appraise time differences over the study period. (I acknowledge the that the authors have provided the conversions in Results section.

– Highlighting the significant predictors in the table(s) would be of great value to readers to follow easier the authors in text description/narrative.

“Specific”

Title: comments/suggestions already stated in “Throughout the text”, and maybe adding the period of study would better elucidate readers, i.e., “…: A 5-year prospective cohort study

Abstract: Good, apart from my comments/suggestions already stated in “Throughout the text”

Keywords: Good, apart from my comments/suggestions already stated in “Throughout the text”

Introduction:

Lines 41-43 – I suggest providing the magnitude of association, “up to...” so that the reader can appraise how correlated those factors were in previous studies.

Methods:

Line 55 – State the country, please

Figure 1 – Shouldn’t this arrow and box be below, i.e., between “Returned survey box and participants for T1”?

Line 88: Why was age categorized? It should be reported. Statistical convenience? Biological or social characteristics throughout those age ranges?

Lines 95-97 – The reasons why should be presented so that it could be appraised by readers

line 101 and throughout – not sure what an health exposure variable means… Can the authors please explain?

lines 148-150 – And what is the criterion/criteria used for decision?

Table 3 – Abbreviations are repeated. Also, see my “Throughout the text” suggestions.

lines 208-214 – effect size statistics should be presented and method and criteria used for decision reported in the statistical analysis section

Discussion:

line 266 – Either a sentence is missing or “protective” is misplaced or a typo.

line 268 – “… has negative effect on unhealthy behaviors.” This appears to be positive, but the way it is written sounds like a bad thing. Can the authors please rephrase?

Lines 297-298: – Are there other instruments with potential to capture this outcome or should be developed? What are the recommendations of the authors based on their findings and experience with IPAQ-L?

Conclusion: Good

Author Response

1.1.          “Throughout the text”:

– The authors omit in most parts of the text the attribute of the sitting term which is sitting time or time spent in sitting. This is relevant because the authors claim they will study sitting patterns, but in fact they only studied overall time spent in sitting, as a measure of sedentary behavior. Sitting patterns could be sitting posture, sitting time or posture while working, sitting time with legs crossed while watching TV, sitting lumbopelvic angle while driving in urban areas, etc. IPAQ-L cannot differentiate between those, hence, “sitting patterns” is not conceptually correct (In this study) and should be amended to sitting time or time spent in sitting whenever applicable.

– I usually don’t recommend changing, in most revisions, the decimals in central tendency and dispersion descriptive statistics in variables related with time that have a particular subdivision system (years and months; hours and minutes), e.g., Age = 54.6 ± 14.3 years old, but because the focus of this research is time spent in sitting, I found that reporting time spent in sitting as 40.2 hours/week can be exhausting for readers to convert constantly the decimals to minutes in order to appraise time differences over the study period. (I acknowledge the that the authors have provided the conversions in Results section).

Highlighting the significant predictors in the table(s) would be of great value to readers to follow easier the authors in text description/narrative.

Response: As per the reviewer’s suggestion we have now replaced “sitting” with sitting time or time spent in sitting as a measurement of sedentary behaviour throughout the text.

Response: We have provided conversion in the results section to enhance readability.

Further, we have now bolded all p- values of significant predictors in Table 3 as per reviewer’s request.

Specific

  • Title: comments/suggestions already stated in “Throughout the text”, and maybe adding the period of study would better elucidate readers, i.e., “…: A 5-year prospective cohort study

Response: We have added number of years (5-years) in the title as suggested.

Abstract: Good, apart from my comments/suggestions already stated in “Throughout the text”

Keywords: Good, apart from my comments/suggestions already stated in “Throughout the text”

Introduction

  • Lines 41-43 – I suggest providing the magnitude of association, “up to...” so that the reader can appraise how correlated those factors were in previous studies.

 Response: We have amended this text which now reads: line 44-47

“For example, studies have found positive associations (higher SEP spent 60-70 min/day more time on sitting than lower SEP ) as well as inverse association (lower SEP spent (70 to 80%) more time on sitting than higher SEP during weekend and leisure) between indicators of SEP and sitting time.”

Methods

  • Line 55 – State the country, please

Response:  We have now stated the country name as suggested (Australia), line 60

  • Figure 1 – Shouldn’t this arrow and box be below, i.e., between “Returned survey box and participants for T1”?

Response: We have now changed the arrow position in flow chart.

  • Line 88: Why was age categorized? It should be reported. Statistical convenience? Biological or social characteristics throughout those age ranges?

Response: We have now described this in the text, ,which now reads:

“Age was categorized (18-25 years, 26-35 years, 36-45 years) as per previous studies have reported that being young (18-25 years) was associated with higher sitting compared to being older.”, lines 94-96.

  • Lines 95-97 – The reasons why should be presented so that it could be appraised by readers

Response: We have now provided the reason why baseline characteristics of participants lost to follow-up should be compared to participants who remained in the study. This text now reads:

“Baseline characteristics of participants lost to follow-up were compared to the participants who completed surveys at all three-time points because difference between the samples could affect the internal validity of the study results.”, lines 102-105

  • line 101 and throughout – not sure what an health exposure variable means… Can the authors please explain?

Response: BMI and self-reported health status are the health exposure variables were which we reported in brackets, lines 109-110

  • lines 148-150 – And what is the criterion/criteria used for decision?

Response: First sensitivity analysis was to compare between the complete case analysis and imputed data to find the impact of missing data. No further criteria were used to for this decision.

Second sensitivity analysis was based on a hypothesis (We hypothesized that participants with high values may have reported on total weekday and weekend estimate rather on single day estimate of weekday and weekend). We have outlined this on line 150-153. Results from both the data sets (data set built based on our hypothesis and data set with high implausible values as missing) were compared to prove our hypothesis.

  • Table 3 – Abbreviations are repeated. Also, see my “Throughout the text” suggestions.

Response: Repeated words are now removed, and p-values of significant predictors have since been edited (bold)

  • lines 208-214 – effect size statistics should be presented and method and criteria used for decision reported in the statistical analysis section

Response: We provided a range of differences between beta values of complete case analysis and imputed data (beta value from complete case analysis – beta value from imputed data) which now reads as:

Sensitivity analyses using complete case data indicated small changes in estimates (difference in beta values range between 0 to 1.2), but results remained consistent and no changes in inferences were apparent (Supplementary Table.2), line 209

Discussion

  • line266 – Either a sentence is missing or “protective” is misplaced or a typo.

Response: Word “protective” was an error and is now removed.

  • line 268 – “… has negative effect on unhealthy behaviors.” This appears to be positive, but the way it is written sounds like a bad thing. Can the authors please rephrase?

Response: We rephrased the sentence which reads now:

“This finding is consistent with existing hypotheses where marriage appears to be protective against unhealthy behaviour.” , lines 267-269

  • Lines 297-298: – Are there other instruments with potential to capture this outcome or should be developed? What are the recommendations of the authors based on their findings and experience with IPAQ-L?

 Response:  Most self-reported questionnaires on sitting time show acceptable measurement properties (test-retest period range from 3 days to 2 months) for establishing cross-sectional association with health outcomes, but they may not necessarily be reliable for assessing the change over longer periods of time (1-4). Further development of reliable measures of sitting, capturing domain specific behaviour is needed.

Conclusion: Good

Reviewer 2 Report

Dear Authors,

  • please specify better, also in the legend what do you mean, in table 1, by General Health N (%) - Excellent-Very good- Good: which parameter did you use to define the state of health in this sense?
  • Also, please explain better which cut of you have given yourself to define a very sedentary lifestyle compared to the cut offs you can define for other lifestyle attitudes

    - Also, in your opinion, is it possible to define a synthesis algorithm of sitting among women over time with respect to patterns and predictors: ? it would be very impactful for your paper.

Author Response

Dear Authors,

  • please specify better, also in the legend what do you mean, in table 1, by General Health N (%) - Excellent-Very good- Good: which parameter did you use to define the state of health in this sense?

Response: The measure of health status is described in section 2.2.2. We have now included further description in the manuscript of the question used, as follows:

“Participants on self-reported health status were asked to respond to the following: “Would you say your health is: excellent, very good, good, fair, poor”, lines 93-94

  • Also, please explain better which cut of you have given yourself to define a very sedentary lifestyle compared to the cut offs you can define for other lifestyle attitudes

Response: We are unsure what cut-off the reviewer is referring to, as we have not used any cut-offs to define very sedentary lifestyles.

  • Also, in your opinion, is it possible to define a synthesis algorithm of sitting among women over time with respect to patterns and predictors: ? it would be very impactful for your paper.

Reviewer 3 Report

Title:

“disadvantaged women” concept should be included in the title.

Abstract

Women's age (range and mean) is not reported.

Socio-demographic and health variables should be reported in methods.

“Mean baseline sitting was 40.9 hrs/week, decreasing to 40.1 hrs/week over five years” This sentence has just to be included if this difference is signficative.

“Annually, the aver-16 age sitting decreased by 0.5 hours/week (95% CI; -0.8 to -0.1) in women working full-time but in-17 creased by 0.1 hours/week…” In Methods is indicated that sitting time is registered three times over 5 years; but in this sentences seems that is registered annually.

Manuscripts

Inclusion and exclusion criterial are not included.

Line 154 indicate that sitting time decrease, however this sentences just is correct if this differences is signiticative, although the statistic is no included.

The age classification on the basis of which concepts it is made. Include this classification in method and include citation.

There is a lack of statistics to support this sentence “Mean sitting time was typically highest amongst youngest participants, living with obesity, living in urban areas, with poor/fair health, higher education, working full-time, higher income, never married and with no children”

Table 2 do not contain any differences statistics

The result for unadjusted modal is described in the result section, however the important result is by adjusted model. Change this information.

In statistical analysis section should be included the covariables used by adjusted model; and the selection of these have to be justification. The use of diferents covariables by adjusted model is confusing and more explaining have to be considered.

Line 182 indicate “sitting time per year”. The units register is hours/weeks.

After read the results, I do not know if the discussion is in base an unadjusted or in base adjusted model result. Please, rewrite in base adjusted model results if it is necessary.

The loosed to follow-up is very high. This aspect has to be comment as a limitation; and an explication is needed.

The usability of this data is no presented in the discussion. Discussion is focus in show the result and comment with other similar studies, but the clinical or social relevance is no indicate or highlight.

The conclusion is weak. It should be improved.

Author Response

Comments and Suggestions for Authors

Title:

3.1        “disadvantaged women” concept should be included in the title.

Response: We have included the word “disadvantaged”  in the title as suggested which now reads:

Patterns and predictors of sitting time among women from disadvantaged neighbourhoods over time : A 5-year prospective cohort study

Abstract:

3.2        Women's age (range and mean) is not reported. Socio-demographic and health variables should be reported in methods.

Response: We have now provided women’s age range and mean (SD) as suggested, line 12. It was not possible to report all exposures in the abstract due to word limits so provided one example of each sociodemographic and health variable, however, detailed descriptions were given of  sociodemographic and health exposures in section 2.2.2.

3.3        “Mean baseline sitting was 40.9 hrs/week, decreasing to 40.1 hrs/week over five years” This sentence has just to be included if this difference is signficative.

 Response: Given this is a descriptive study, we included this information for descriptive purposes, while statistical tests are included in our longitudinal models of change.

3.4        “Annually, the aver-16 age sitting decreased by 0.5 hours/week (95% CI; -0.8 to -0.1) in women working full-time but in-17 creased by 0.1 hours/week…” In Methods is indicated that sitting time is registered three times over 5 years; but in this sentences seems that is registered annually.

Manuscripts:

3.5        Inclusion and exclusion criterial are not included.

Response: Details of our sample was defined in section 2.1 and presented in a flow chart   (figure. 1). Ineligible women’s details were described in one box of the flow chart.

3.6        Line 154 indicate that sitting time decrease, however this sentences just is correct if this differences is signiticative, although the statistic is no included.

Response: Please see response to reviewer comment 3.3 above.

3.7        The age classification on the basis of which concepts it is made. Include this classification in method and include citation.

Response: Please see response to reviewer comment 1.6

3.8        There is a lack of statistics to support this sentence “Mean sitting time was typically highest amongst youngest participants, living with obesity, living in urban areas, with poor/fair health, higher education, working full-time, higher income, never married and with no children”

Response: These are descriptive and therefore p-values are unnecessary. We have not said there are statistically significant differences so there is no error here.

3.9        Table 2 do not contain any differences statistics

Response: These are descriptive of a larger sample (n=4349). We have only described the sample, not stated the difference between the groups based on statistical significance.

3.10     The result for unadjusted modal is described in the result section, however the important result is by adjusted model. Change this information.

Response: We have now reported the results of adjusted models in result section (3.4) as per suggested, lines 190-205

3.11     In statistical analysis section should be included the covariables used by adjusted model; and the selection of these have to be justification. The use of diferents covariables by adjusted model is confusing and more explaining have to be considered.

Response: We described the criteria for selecting covariates  for different models in our statistical analysis section (2.3) line 121-22 with reference, as follows:

 ‘Adjusted models used the criteria of a 10% change in estimates to identify confounders for each individual model.’

Also, in the footnote of table no.3 we provided superscript (a-j) for each exposure variable in the table.

3.12     Line 182 indicate “sitting time per year”. The units register is hours/weeks.

Response: We calculated the annual change for sitting time in hours/week. To clarify this, we have now included the following text on line 190 and 196 which now reads:

“In regards to the annual change in setting time, in adjusted models for employment status, annually sitting time decreased on average by 0.4 hours/week (95% CI: -0.7 to -0.05), or 24 mins/week.”

In regard to the annual change in sitting time in the adjusted model for number of children, annually sitting time decreased on average by 0.6 hours/week (95% CI: -0.2 to 1.3), or 36 mins/week.

3.13     After read the results, I do not know if the discussion is in base an unadjusted or in base adjusted model result. Please, rewrite in base adjusted model results if it is necessary.

Response: Discussion on was written on the basis of adjusted results. We have now included the following text to clarify, line 219

“Based on the findings of adjusted models, in this sample, sitting time declined over time, identical to previously reported studies on European and Australian adults.”

3.14     The loosed to follow-up is very high. This aspect has to be comment as a limitation; and an explication is needed.

Response: We addressed loss to follow-up as a limitation  (section 4.2, lines 300-302) as well as used multiple imputation to decrease the biasness.

3.15     The usability of this data is no presented in the discussion. Discussion is focus in show the result and comment with other similar studies, but the clinical or social relevance is no indicate or highlight.

Response: We have highlighted the social relevance of findings by providing potential explanations of each key finding. This can be found for example on lines 260-263 and 281-88

e.g., “As mid-aged women are more likely to have older children with fewer physical de-mands than required for younger children (e.g., less time in childcaring activities), this may result in increased leisure-time sitting, such as watching TV. Alternatively, mothers with two or more children may spend more time transporting their children in cars (e.g., school, extra-curricular activities) and sitting watching them participate in various activities or may have greater work hours.”

3.16     The conclusion is weak. It should be improved.

Response: We have now strengthened the conclusion, lines 309-316  as follows:

“This study identified two population subgroups (those not working and those with two or more children) of women from socioeconomically disadvantaged neighbourhoods at greatest risk of increasing sitting time. Given the detrimental health effects of sitting time, these groups should be targeted in future interventions for reducing SB. Furthermore, for a better understanding of potential factors for these associations and inform development of SB interventions for these groups, future studies need to investigate the predictors associated with change in domain (e.g., leisure, transport, work) and context- (e.g., TV viewing, computer use) specific sitting behaviors.”
